

# Application of fractal models to delineate mineralized zones in
# the Pulang porphyry copper deposit, Yunnan, Southwest China
Xiaochen Wang[a], Qinglin Xia[a,b,*], Tongfei Li[a], Shuai Leng[a], Yanling Li[a],
Li Kang[a], Zhijun Chen[a], Lianrong Wu[c]
[a] Faculty of Earth Resources, China University of Geosciences, Wuhan 430074, China
[b] Collaborative Innovation Center for Exploration of Strategic Mineral Resources,
Wuhan 430074, China
[c] Yunnan Diqing Nonferrous MetalCo., Ltd., Shangri-La674400, China
Abstract
The purpose of the paper is to depict various mineralized zones and the barren
host rock in accordance with the subsurface and surface lithogeochemical data using
the concentration–volume (C–V) and power spectrum–volume (S–V) fractal models
within the Pulang copper deposit, southwest China. Results obtained by
concentration–volume model depict four geochemical zones defined by Cu thresholds
of 0.25%, 1.38% and 1.88%, which represent non-mineralized wall rocks (Cu<0.25%),
weakly mineralized zones (0.25%–1.38%), moderately mineralized zones
(1.38%–1.88%), and highly mineralized zones (Cu>1.88%).S–V model is utilized by
performing 3D fast Fourier transformation for assay data in the frequency domain.
The S–V method indicates three mineralized zones characterized by Cu threshold
valuesof 0.23% and 1.33%. The zones of <0.23% Cu represent barren host rocks and
zonesof 0.23%-1.33% Cu represent the hypogene zones and zones >1.33% Cu
represent supergene enrichment zones. Both the multifractal models show that high
grade mineralization is located at the center and south of Pulang deposit. The results
are in contrast with alteration and mineralogical models resulted from the 3D geologic
model utilizing the logratio matrix method. Better results were obtained from S–V
model to delineate high grade mineralization of Pulang deposit.However, results of
C–V method of moderate and weak grade mineralization are more precise than the
results gained from S–V method.



Keywords:Fractal; Concentration–volume model (C–V); Power spectrum–volume
model (S–V); Mineralized zone; the Pulang porphyry copper deposit

## 1. Introduction

The depiction and recognition of various mineralized zones and barren host rock
is the primary goal of the mineral exploration work. The research of
systematic ore-forming mineralogy offers helpful data about the metallogenic
processes of deposits, for the mineral assemblages of different types of deposits
reflect the typical characteristics (White and Hedenquist, 1995; Craig andVaughan,
1994).Common means are on the basis of mineralography, petrography and alteration
minerals assemblages to delineate various mineralized zones in porphyry deposits
(Beane, 1982; Schwartz, 1947;Sillitoe, 1997; Berger et al., 2008). Lowell(1968)
firstly put forward a theory model which indicated the mineralogy variations of lateral
and vertical directions in the alteration zones. Some comparable models are usually
proposed related to potassic zones frequently situated in the center and deep of
porphyry ore deposits on the basis of this model (Sillitoe and Gappe, 1984; Cox and
Singer, 1986;Melfos et al., 2002).There are also other methods such as stable isotope
studies and fluid inclusion to outline various mineralization phases(Boyce et al., 2007;
Wilson et al., 2007). The drillhole data with logging information containing
mineralographical information, host rock changes and alteration is helpful to delineate
the mineralization zones. The boundaries of different zones can be exhibited by
different geological interpretations and various results can be obtained.
Non-Euclidian fractal geometry is an significant branch of non-linear sciences. It
is utilized in various research fields of geosciences since 1980s (Mandelbrot,
1983).The correlations between geology, geochemistry and mineralogical
backgrounds with spatial information can be researched by the methods on the basis
of fractal geometry (Carranza,2008, 2009). The fact that the fractal dimensions exist
in different geochemical patterns of diverse elements has been shown by Bolviken et
al. (1992) and Cheng et al. (1994). The concentration–area(C–A) fractal method was
put forward by Cheng et al. (1994) to recognize geochemical anomalies from



backgrounds and calculate thresholds of geochemical data of different elements.
Furthermore, there are many other fractal methods proposed and utilized in
exploration work of geochemistry including number–size (N–S) fractal method
proposed by Mandelbrot (1983), concentration–perimeter(C–P) fractal method
proposed by Cheng (1995), power spectrum–area(S–A) fractal method proposed by
Cheng et al.(1999), concentration–distance (C–D)fractal method proposed by Li et
al.(2003), concentration–volume (C–V) fractal method proposed by Afzal et al.(2011)
and power spectrum–volume (S–V) fractal method proposed by Afzal et al.(2012).
Different geochemical processes could be described by the diversities within fractal
dimensions, which obtained by research of relative geochemical data. Afzal et
al.(2011) considered that the log–log plots obtained by fractal methods are useful
means to delineate different populations of geochemical data and the thresholds could
be determined as some break points in plots.
The utilization of fractal models to delineate various grade mineralization is
dependent on the correlations of metal grades and volumes (Afzal et al., 2011; Cheng,
2007; Simet al., 1999; Agterberg et al., 1993). The concentration–volume (C–V) and
power spectrum–volume (S–V) fractal methods were put forward by Afzal et al. (2011,
2012) to delineate various grade mineralization. We utilized C–V and S–V fractal
methods to delineate diverse mineralized zones and host rocks of Pulang copper
deposit within this paper.
**2. Fractal models**
2.1. Concentration–volume fractal model
Afzal et al. put forward concentration–volume fractal method in 2011 based on
the same principle of the concentration–area method (Cheng et al., 1994) in order to
analysis the correlation between the concentration of ore elements and relevant
occupied volume which its concentration is above or less than the presented value
(Afzal et al., 2011;Sadeghi et al., 2012; Soltani et al.,2014; Zuo et al., 2016).It could
be shown as:
$\qquad V(\rho \leq \upsilon) \propto \rho^{-a_1}; \ V(\rho \geq \upsilon) \propto \rho^{-a_2}$ \hfill (1)



V(ρ≥υ) and V(ρ≤υ) represent those occupied volumes which concentrations are above
or equal to and less than or equal to presented value υ; υ indicates the threshold
between two zones; $a_1$ and $a_2$ indicate the characteristic indexes.Thresholds obtained
by this method indicate the boundaries of diverse grade mineralization of ore deposits.
The drill hole data of elemental concentration values are interpolated with the method
of geostatistical estimation to compute V(ρ≥υ) and V(ρ≤υ).They are those volume
values surrounded with the given value υ within a 3D model.
## 2.2. Power spectrum–volumefractal model
Different geochemical patterns existed within spatial domain could be seen as
layered signals with various frequencies.Cheng et al. (1999) put forward power
spectrum–area fractal method to recognize geochemical anomalies from backgrounds
utilizing the method of spectrum analysis within frequency domain according to this
argument. This model is combined with concentration–area method (Cheng et al.
1994). It offers an useful mean to determine an optimum threshold value between
various forms based on different scaling property.
Afzal et al.(2012) put forward power spectrum–volume (S–V) fractal method to
delineate different grade mineralization based on the same idea as the S–A method
proposed by Cheng et al.(1999).S–V method was utilized in frequency domain. And it
was performed by applying the fast Fourier transformation for assay data. The straight
lines obtained by log–log plots indicate the relationships between power spectrums
and relative volumes of ore elements. They were utilized to recognize the hypogene
zones and supergene zones from barren host rocks and leached zone of the deposit.
The recognization of various mineralization zones is on the basis of the power–law
correlations of power spectrums and relative volumes. The formula is as follows:
$V(\geq S) \propto S^{-2/\beta}$              (2)
Where, the relationships of power spectrums (S=−‖F(Wx, Wy, Wz)‖) and
occupied volumes which power spectrums are greater than or equal to S can be
indicated by this form; F represents the fast Fourier transformation for the
measurement μ(x, y, z); Wx, Wy and Wz seperately indicate wavenumbers or angular





frequencies of the directions of X, Y and Z axis of a 3D model.The range of index β is
0<β≤2 or 1≤2/β with particular circumstance of β=2 or 2/β=1 related to monofractal
or non-fractal and 1<2/β to multifractal (Cheng, 2006).
By utilizing the method of geostatistical estimation, drill hole data of elemental
concentration values are interpolated to construct the block model with ore element
distribution. The power spectrum values can be obtained by utilizing 3D fast Fourier
transformation for ore element grades.
The obtained data was classified to a number of classes. The determination of the
amount of classes should consider the gross amount of data at a required precise level.
The range value from the minimum to maximum values of power spectrum was
calculated and the width of each class was finally decided by separating the range into
the amount of classes. Then we count the amount of voxels of each class and compute
their accumulative volume values. And all of the considered voxels are counted as
points because they have constant volumes. The logarithm of all power spectrum
values and accumulative volume values were calculated. And the log-log plot of
power spectrums and volumes was drawn according to previous counted values.
Then the filters were constructed on the basis of threshold values obtained by the
log-log plot of S-V. Finally, the resulted power spectrums were converted back to
space domain by utilizing inverse fast Fourier transformation.
**3. The geological setting of Pulang copper deposit**
The Pulang depositis situated in the southern end of Yidun continental arc of
southwest China (Fig. 1). The continental arc is generated due to the westward
subduction of Garze–Litang oceanic crust(Deng et al., 2014b, 2015; Wang et al.,
2014). And Leng et al. (2012) and Li et al.(2011, 2013) have systematically
researched detailed geological characteristics of Pulang deposit, such as the
representative porphyry alteration zones, the geometry of orebody, metallogenic time
and the geodynamic settings of this deposit.The Pulang deposit consists of five
ore–bearing porphyries. They cover an range of about 9 square kilometers. Liu et al.
(2013) showed that Cu ore tonnage of Pulang deposit is reckoned to be 6.50 Mt.





The outcrop strata of Pulang deposit mainly consist of clastic rocks, andesite and quaternary sediments of UpperTriassic Tumugou Formation (Fig.1c).The Triassic porphyry intrusions mainly comprise quartz monzonite porphyry, quartz diorite porphyry,quartz diorite porphyrite and granodiorite porphyry. The Tumugou Formation strata was intruded by the quartz diorite porphyry with an age of $219.6 \pm 3.5$ Ma obtained by Zircon U–Pb dating(Pang et al., 2009). Then quartz monzonite porphyry with an age of $212.8 \pm 1.9$ Ma and granodiorite porphyry with an age of $206.3 \pm 0.7$ Ma obtained by Zircon U–Pb dating (Liu et al., 2013) seperately crosscut quartz diorite porphyry.The quartz monzonite porphyry is related to mineralization for its age is similar with the Re–Os isochron age of $213 \pm 3.8$ Ma from molybdenite of deposit (Zeng et al., 2004). Moreover, the Cu grades of quartz monzonite porphyry are higher than the other porphyries.

<Fig. 1 inserts here>

The porphyry-type alteration zones transform from potassium–silication, quartz–sericitization to propylitization zones upward and outward from the center of quartz monzonite porphyry(Fig.4).Most country rocks close to the porphyries were transformed to hornfels. The fact that potassic and quartz–sericitization zones control most orebodies has been validated by the systematic drilling. They constitute the core of mineralized zones. And the weak mineralization often appear in the propylitic zones and hornfels surrounding the core.The orebodies occur as veins within the propylitic zones and hornfels.Major rock types in the deposit are quartz monzonite porphyry, quartz diorite porphyrite, granite diorite porphyry, quartz diorite porphyry and hornfels(Fig. 2). Metallic minerals mainly include chalcopyrite, pyrite and some molybdenite and pyrrhotite (Fig. 3).

<Fig. 2 inserts here>

<Fig. 3 inserts here>

<Fig. 4 inserts here>

## 4. Fractal modeling





On the basis of the geological data of this deposit, such as the collar coordinates,
azimuth, dip, mineralogy and lithology of 130 drill holes, 19996 samples were
gathered from these drill holes every other 2 meters. The laboratory of the 3rd
geological team of Geology and Mineral Resources Bureau of YunnanDiqing
Nonferrous Metal Co. Ltd. utilized iodine–fluorine and oscillo-polarographic method
to analyze the concentrations of Cu and associated paragenetic elements of all the drill
holes and its analytical uncertainty is less than 7%. Only Cu concentrations were
researched in this study. The distribution of Cu concentrations is presented in Fig. 5
with Cu mean value of 0.296%. The experimental semi–variogram of Cu data of
Pulang deposit indicates that these values of the nugget effect and range are 0.126 and
160.0m, seperately(Fig. 6).The spherical model is fitted in regard to the experimental
semi–variogram.The 3D model of Cu concentrations dispersion of Pulang deposit is
produced by utilizing ordinary kriging method of the Geovia Surpac on the basis of
the semi–variogram and anisotropic ellipsoid. Goovaerts (1997) showed that the
values in un-sampled locations are estimated by the ordinary kriging method
according to moving average of interest variables fitting various distribution patterns
of data.It is a spatial estimation means and its error variance related to characteristics
and patterns of the data is minimized. The obtained block model by this method are
utilized as input to fractal models.The Pulang deposit is modeled by 20m $\times$ 20m $\times$ 5m
voxels and they are decided by the grid drilling dimensions and geometrical
characteristics of the Pulang deposit (David, 1970). Pulang deposit is totally modeled
with 150,973 voxels. Different mineralized zones are classified on the basis of these
two fractal methods in this deposit.

<Fig. 5 inserts here>

<Fig. 6 inserts here>

## 4.1. Concentration–volume (C–V) fractal modeling

The occupied volume values related to Cu grades are computed to obtain the
concentration–volume model according to the 3D model of Pulang deposit.Through
the obtained log–log plot of concentrations vs volumes, the threshold values of Cu



grades were determined (Fig.7). It indicates the power-law relation of Cu grades and
volumes. Three thresholds and four populations are gained from C–V log–log plot,
consequently. The first Cu threshold is 0.25%. The range of Cu values of <0.25%
represent barren host rock.The second Cu threshold is 1.38%, and values of
0.25–1.38% Cu represent weak grade mineralization.And the third Cu threshold is
1.88%. The range of Cu values of 1.38–1.88% denote moderately mineralized zones,
and values of >1.88% Cu indicate highly mineralized zones (Table 1). According to
the results, the low concentration zones develop in a lot of sections of Pulang deposit
and are inclined to the northwest–southeast direction of the deposit. Moderately and
highly mineralized zones are located at several parts of the center and south of Pulang
deposit(Fig. 8).

<Fig. 7 inserts here>

<Fig. 8 inserts here>

< Table 1 inserts here>

## 4.2. Power spectrum–volume (S–V) fractal modeling

According to the geological data from this deposit, such as the collar coordinates,
azimuth, dip, mineralogy and lithology of 130 drill holes, a 3D model and block
model of Cu grades dispersion of Pulang deposit were constructed by ordinary kriging
method utilizing the Geovia Surpac.
The power spectrum (S) of Cu grades distribution are computed by utilizing 3D
fast Fourier transformation by MATLAB (R2016a). The logarithmic values of power
spectrum and relevant volume values are fitted against each other (Fig. 9). The
straight lines fitted through the log–log plot indicate the relation of power spectrums
and occupied volumes. The results have indicated that there are two thresholds and
three populations. The thresholds of logS=7.81 and logS=8.70 are decided by the
log–log S–V plot. The 3D filters were designed to separate different mineralization
zones on the basis of these threshold values. Inverse fast Fourier transformation was
utilized to convert the resulted power spectrums back into space domain by MATLAB
(R2016a). According to the results, Cu grades of hypogene zones range from 0.23% to
1.33% (Table 2), and values of >1.33% Cu refer to the supergene enrichment zones,



whereas values of <0.23% Cu pertain to the leached zone and barren host rock(Fig.

10).

<Fig. 9 inserts here>
<Fig. 10 inserts here>
< Table 2 inserts here>

## 238 5. The contrast of results of fractal models and geologic models of

## 239 Pulangdeposit

Lowell and Guilbert(1970) depicted that the alteration models are very critical
within zone recognition. The potassic and phyllic alterations control the most
mineralization within supergene and hypogene zones according to these models. The
various mineralization zones obtained by the fractal methods could be in contrast with
geologic data to verify these results.
Results of fractal models of Pulang deposit were in contrast with 3D geologic
model of Pulang deposit constructed by utilizing Geovia Surpac and drillholes data
(Fig. 2). Furthermore, results gained from fractal models are also dominated by
mineralogical research.
The analysis of spatial relationships of two binary particularly geology and
mathematics models has been indicated by Carranza (2011).The intersection operation
between the mineralization zones obtained from fractal models and alteration zones is
carried out to derive the amount of voxels related to every class of overlap zones
(Table3). And overall accuracy (OA) values of different grade mineralization obtained
by these fractal methods are in contrast with each other.
The contrast between highly mineralized zones on the basis of the fractal models
and potassic zones resulted from 3D geologic model illustrates that the results of these
two fractal models are similar.The OA values of C–V and S–V methods are 0.50 and
0.52 as shown in Table 4, which illustrate that the S–V model gets more accurate
results to recognize high grade mineralization of Pulang deposit.
The contrast between phyllic alteration zones resulted from 3D geologic model
and moderate grade mineralization obtained from fractal methods indicates that OA



values of C–V and S–V fractal methods in regard to phyllic alteration zones of the
geological model are 0.59 and 0.56 (Table 5). The OA values of moderate and weak
grade mineralization zones gained from C–V model is better than the results gained
by S–V model.
It could be considered that there are spatial correlations between different grade
mineralization and geologic features for instance alterations and mineralogy. Several
samples of drillholes are gathered from different grade mineralization zones of Pulang
deposit to validate the results of fractal models. PL-B82 was collected from supergene
enrichment zones with high chalcopyrite content (Fig.13a). PL-B62 and PL-B74
samples were collected from the hypogene zones with low chalcopyrite content and
some pyrrhotite content, respectively (Fig.13b and Fig.13c). PL-B94 sample was
collected from leached zone and barren host rock with lower and no chalcopyrite
content (Fig.13d).

<Fig. 11 inserts here>

<Fig. 12 inserts here>

<Fig. 13 inserts here>

**6. Conclusions**
This study utilized the concentration–volume(C–V) and power spectrum–volume
(S–V) fractal models to delineate and recognize different grade Cu mineralization of
Pulang copper deposit. Both the fractal models reveal high grade Cu mineralization is
located at the center and south of Pulang deposit.The Cu threshold of high grade
mineralization is 1.88% according to C–V method. And Cu threshold of supergene
enrichment zones is 1.33% on the basis of S–V method. Models of moderate grade
mineralization zones contain 1.38–1.88% Cu due to C–V method. And the hypogene
zones contain 0.23–1.33% Cu according to the S–V model.
The C–V method shows barren host rock includes <0.25% and weak grade
mineralization include 0.25–1.38% Cu. And the S–V model reveals that barren host
rock and leached zone contain <0.23% Cu.
The high grade Cu mineralization determined by fractal methods, specially by





S–V method, give better relations with potassic zones of the 3D geologic model based
on the relationship between results obtained from fractal methods and geologic
logging of drill holes of Pulang deposit. In addition, there is a better correlation of
moderate and weak grade mineralization obtained from C–V method and phyllic
alteration zones based on the 3D geologic model.

< Table 3 inserts here>

< Table 4 inserts here>

< Table 5 inserts here>


## Acknowledgements

This research was supported by the National Key R&D Program of China
(2016YFC0600508). The authors thank Tao Dong, Haijun Yu, Qiwu Shen, Zhipeng Li,
Baosheng Shi and Jinhong Yang for supporting in field investigation and providing
parts of raw data.
















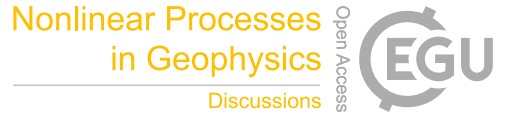

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



**Fig.1.** Geological map of the Pulang porphyry copper deposit, SW China.Modified
after Yunnan Diqing Nonferrous Metal Co. Ltd., 2009.
**Fig.2.** Geological 3D models including lithology, alterationand 3Ddrillholeplot with
the legend of each in thePulang porphyry copper deposit. (Scale is in m$^3$.)
**Fig.3.** Photographs of alteration and mineralization in the Pulang porphyry copper
deposit, SW China. (a) Quartz monzonite porphyry with potassium-silicate alteration;
(b) Quartz diorite porphyrite with quartz-sericite alteration; (c) Quartz diorite
porphyrite with propylitic alteration; (d) Hornfels. Qtz=quartz; Pl=plagioclase;
Kfs=K-feldspar; Bt=biotite; Ser=sericite; Chl=chlorite; Ep=epidote; Py=pyrite;
Ccp=chalcopyrite; Mo=molybdenite; Po= pyrrhotite.
**Fig.4.** Cross section along exploration line 0 in the Pulang porphyry copper deposit,
SW China. Modified after Wang et al., 2012.
**Fig.5.** Histogram of Cu concentrations in lithogeochemical samples from the Pulang
deposit.
**Fig.6.** The experimental semi–variogram (omni-directional) of Cu data in Pulang
deposit.
**Fig.7.** C–V log–log plot for Cu concentrations in the Pulang deposit.
**Fig.8.** Zones in the Pulang deposit based on thresholds defined from the C–V fractal
model of Cu data: (a) highly mineralized zones; (b) moderately mineralized zones; (c)
weakly mineralized zones; (d) barren host rock.(Scale is in m$^3$.)
**Fig.9.** S–V log–log plot for Cu concentrations in the Pulang deposit.
**Fig.10.** Zones in the Pulang deposit based on thresholds defined from the S–V fractal
model of Cu data: (a) the supergene enrichment zones; (b) the hypogene zones; (c) the
leached zone and barren host rock (Scale is in m$^3$.)
**Fig.11.** Highly mineralized zones in the Pulang deposit: (a) potassium-silicate zone
resulted from the 3D geological model from drillcore geological data; (b) C–V
modeling of Cu data; and (c) S–V modeling of Cu data(Scale is in m$^3$.)
**Fig.12.** Moderately mineralized zones in the Pulang deposit:(a) quartz–sericite zones
resulted from the 3D geological model from drillcore geological data; (b) C–V
modeling of Cu data; and (c)S–V modeling of Cu data (Scale is in m$^3$.)
**Fig.13.** Chalcopyrite content in several samples based on mineralographical study: (a)
PL-B82 sample collected from supergene enrichment zones; (b) PL-B62 sample
collected from the hypogene zones; (c) PL-B74 sample collected from the hypogene
zones; (d) PL-B94 sample collected from leached zone and barren host rock.
Po= pyrrhotite; Ccp=chalcopyrite.











**Table 1** Thresholds concentrations obtained by using C–V model based on Cu% in
Pulang deposit.
**Table 2** Ranges of power spectrum (S) for different mineralization zones in Pulang
deposit.
**Table 3** Matrix for comparing performance of fractal modeling results with geological
model. A, B, C, and D represent numbers of voxels in overlaps between classes in the
binary geological model and the binary results of fractal models (Carranza, 2011).
**Table 4** Overall accuracy (OA), Type I and Type II errors (T1E and T2E, respectively)
with respect to potassic alteration zone resulted from geological model and threshold
values of Cu obtained through C–V and S–V fractal modeling.
**Table 5** Overall accuracy (OA), Type I and Type II errors (T1E and T2E, respectively)
with respect to phyllic alteration zone resulted from geological model and threshold
values of Cu obtained through C–V and S–V fractal modeling.







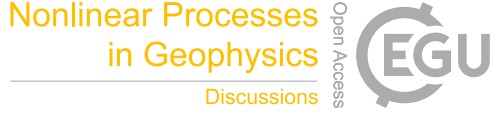

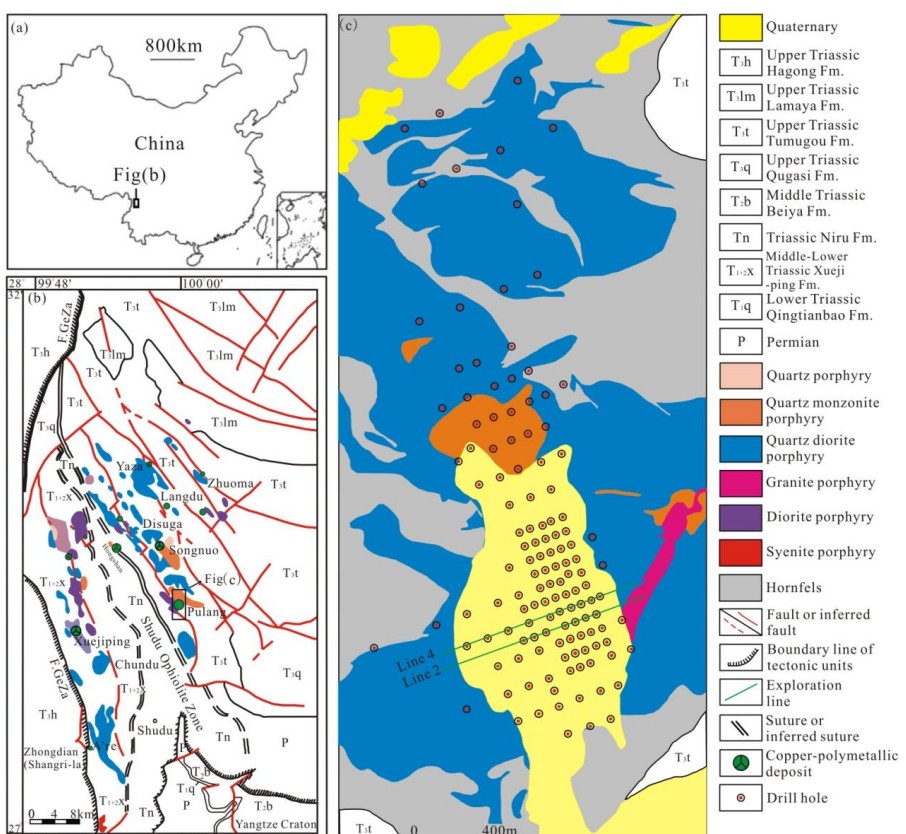

**Fig. 1.**



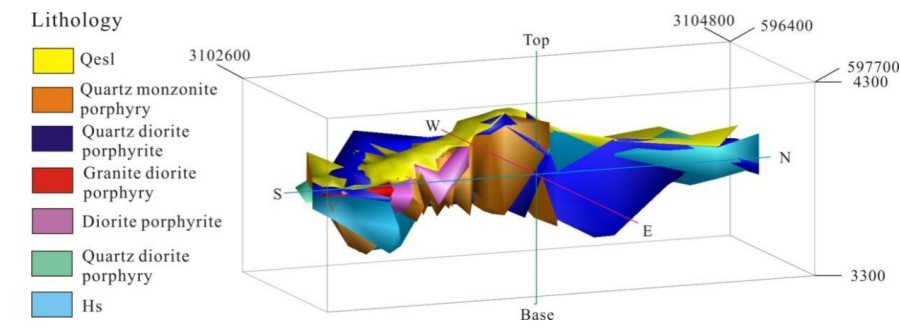

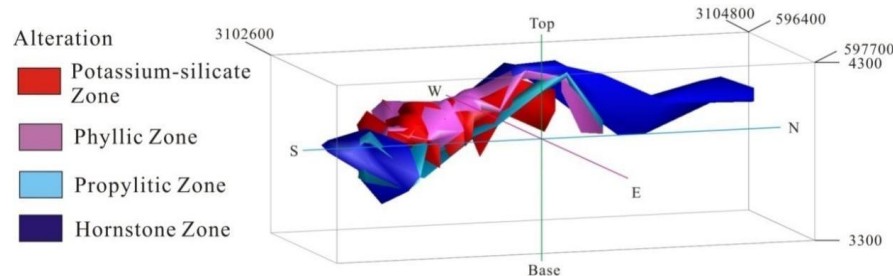

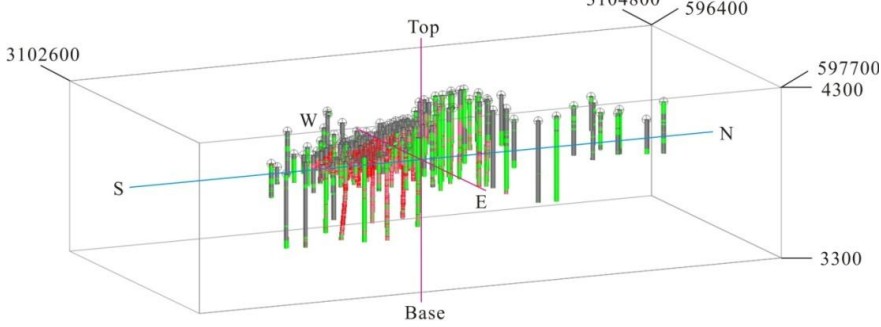

**Fig. 2.**



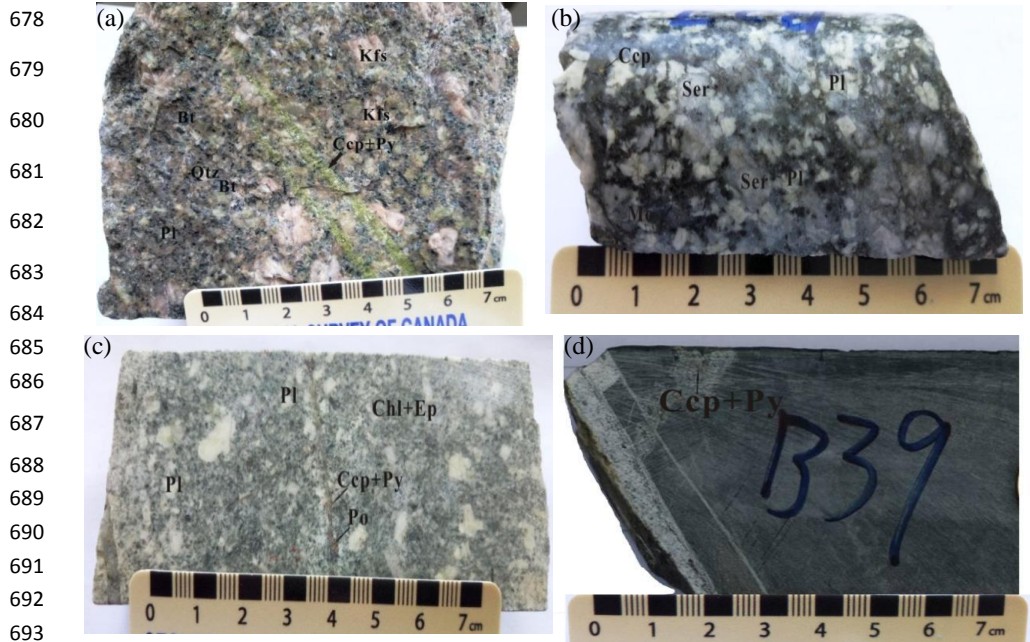

**Fig. 3.**

**Fig. 4.**





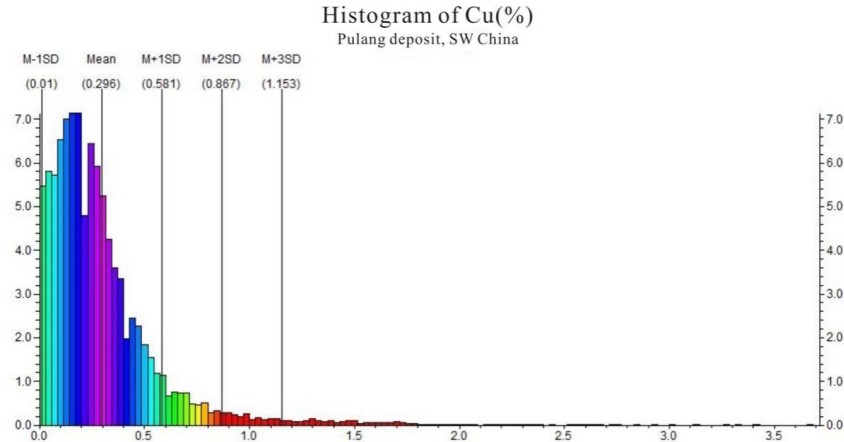


**Fig. 5.**

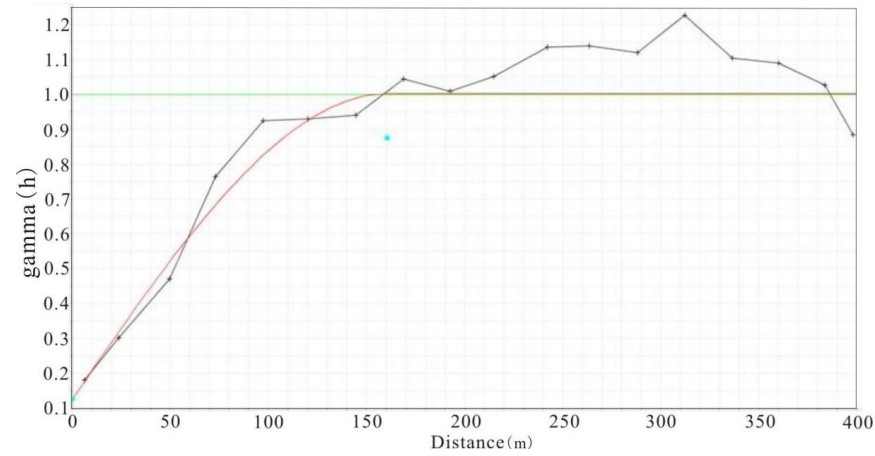


**Fig. 6.**




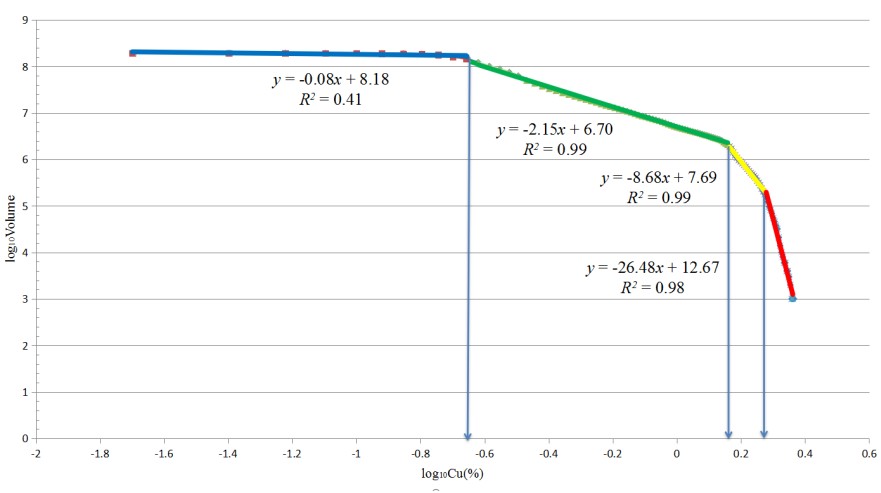

**Fig. 7.**
























(a)

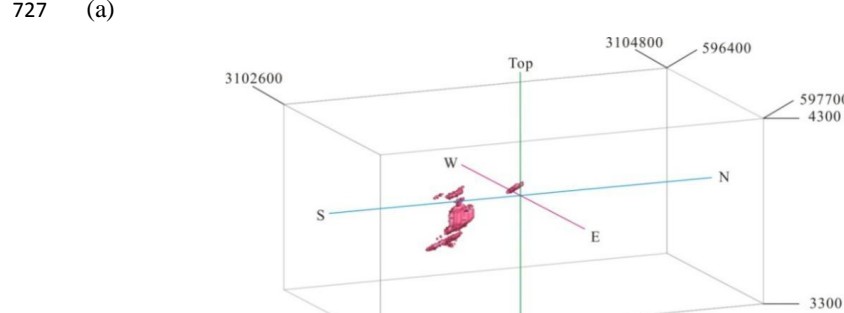


(b)

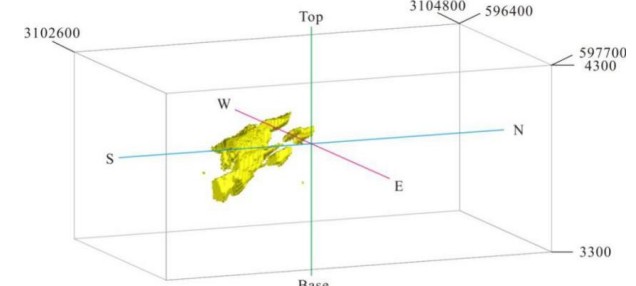


(c)

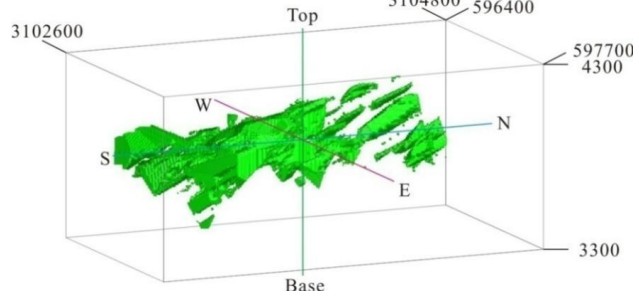


(d)

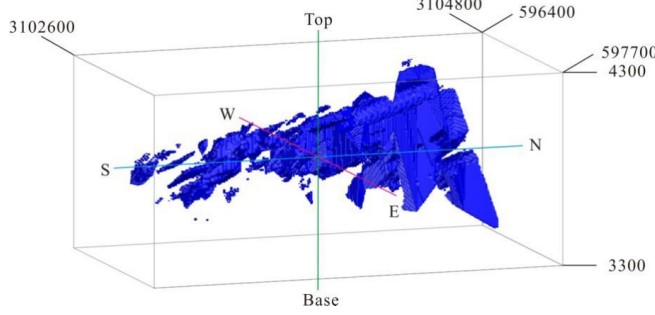


**Fig. 8.**



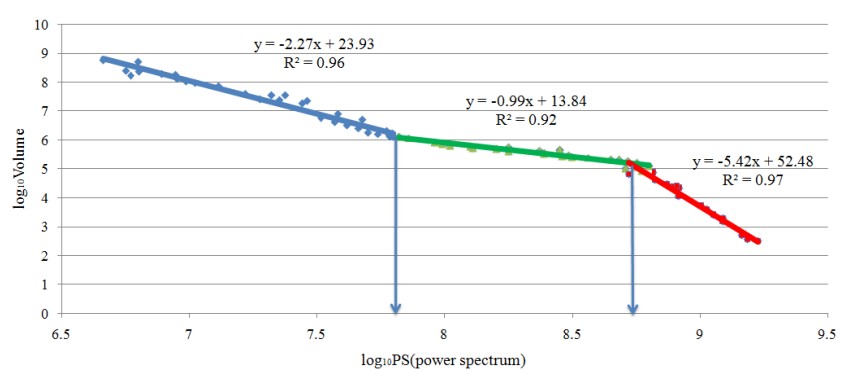


**Fig. 9.**
(a)

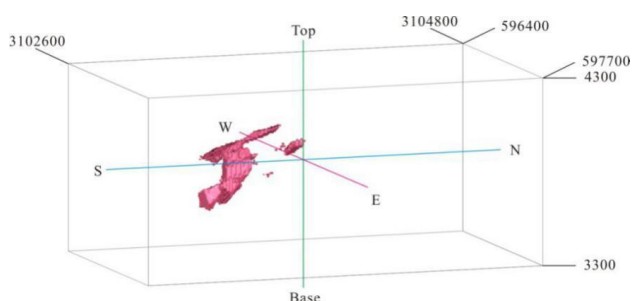


(b)

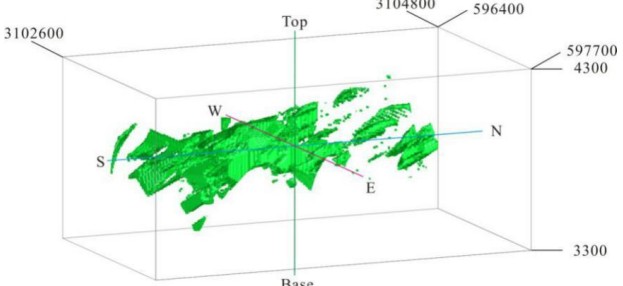


(c)

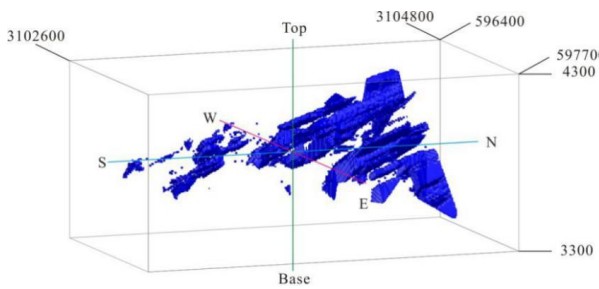


**Fig. 10.**
(a)

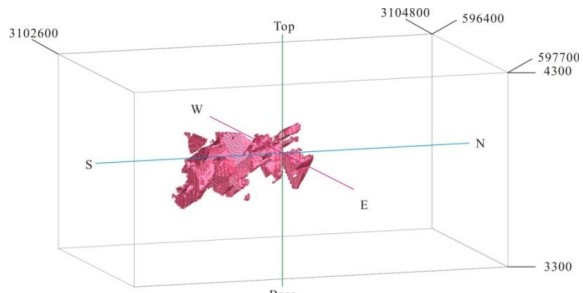

(b)

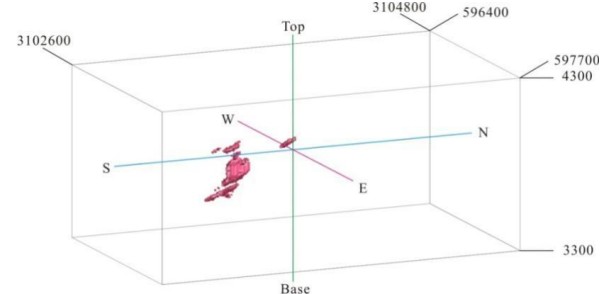

(c)

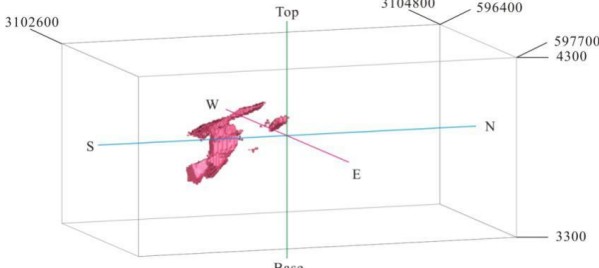


**Fig. 11.**











(a)

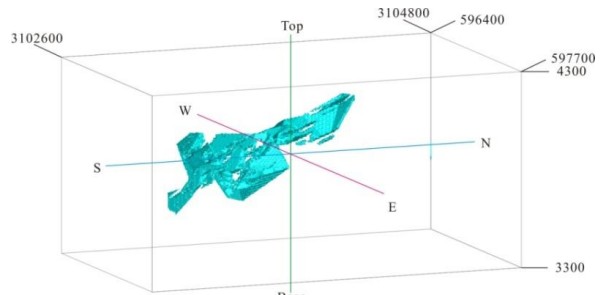


(b)

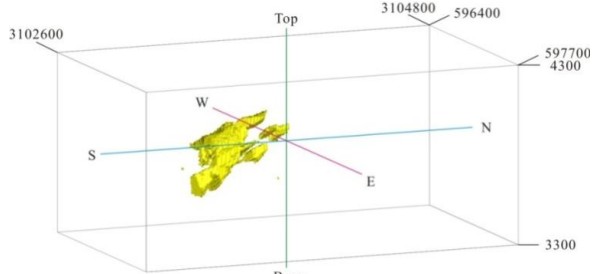


(c)

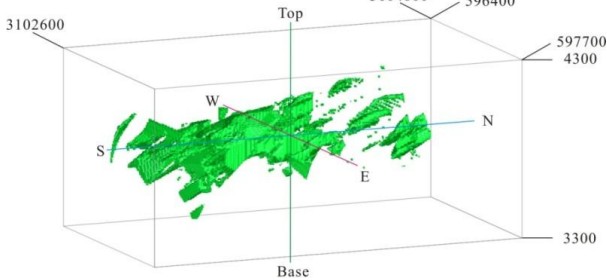


**Fig. 12.**









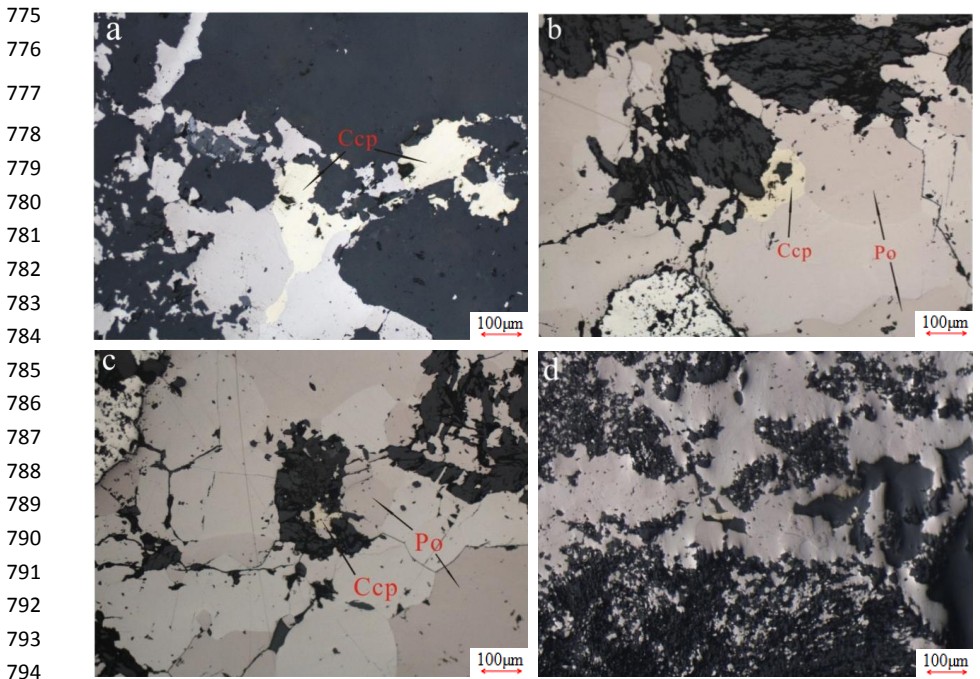

**Fig. 13.**




**Table 1**

| Mineralized zones | Thresholds(Cu%) | Range(Cu%) |
|---|---|---|
| Barren host rock | | <0.25 |
| Weakly mineralized | 0.25 | 0.25–1.38 |
| Moderately mineralized | 1.38 | 1.38–1.88 |
| Highly mineralized | 1.88 | >1.88 |

**Table 2**

| Mineralized zones | PS threshold | Range of PS | Range(Cu%) |
|---|---|---|---|
| leached zone and barren host rock | | <7.81 | <0.23 |
| hypogene zones | 7.81 | 7.81-8.70 | 0.23-1.33 |
| supergene enrichment zones | 8.70 | >8.70 | >1.33 |

**Table 3**

| | | Geological model | |
|---|---|---|---|
| | | Inside zone | Outside zone |
| Fractal model | Inside zone | True positive (A) | False positive (B) |
| | Outside zone | False negative (C) | True negative (D) |
| | | TypeIerror=C/(A+C) | TypeIIerror=B/(B+D) |
| | | Overallaccuracy=(A+D)/(A+B+C+D) | |


**Table 4**

| | | Potassic alteration of geological model | |
|---|---|---|---|
| | | Inside zones | Outside zones |
| C–V fractal model of highly mineralized zones | Inside zones | A  2850 | B  1360 |
| | Outside zones | C  77927 | D  76913 |
| | | T1E  0.96 | T2E  0.02 |
| | | OA | 0.50 |
| S–V fractal model of supergene enrichment zones | Inside zones | A  4131 | B  2318 |
| | Outside zones | C  73985 | D  74726 |
| | | T1E  0.95 | T2E  0.03 |
| | | OA | 0.52 |







**Table 5**

|  |  | Phyllic alteration of geological model | | | |
|  |  | Inside zones | | Outside zones | |
| C–V fractal model of moderately and weakly mineralized zones | Inside zones | A | 36518 | B | 48027 |
|  | Outside zones | C | 25461 | D | 69155 |
|  |  | T1E | 0.41 | T2E | 0.40 |
|  |  | OA | | 0.59 | |
| S–V fractal model of the hypogene zones | Inside zones | A | 40080 | B | 44943 |
|  | Outside zones | C | 26899 | D | 54239 |
|  |  | T1E | 0.40 | T2E | 0.45 |
|  |  | OA | | 0.56 | |
