# Peer review of "Application of fractal models to delineate mineralized zones in"

_Nonlinear Processes in Geophysics, 2019_

## Referee Comment (RC1) · Anonymous Referee #1 · 10 Apr 2019

General comments.

In the paper, the authors apply two methods based on fractal analysis to Cu concentration in order to analyze the mineralized zones of a copper mine. Authors closely follow the logic and the methods described in the rightly referenced articles by Afzal et al. (2011 and 2012) and compare the results obtained from the application of the two procedures. The paper can be interesting for data content and for the comparison made.

Unfortunately, the language is quite poor as it presents some traduction and grammar errors and it is sometimes difficult to follow the logic of the text. Some parts are

rather obscure (e.g. lines 123-124 or 249-253) A revision by a mother-tongue is recommended.

Specific comments.

- The histogram of Cu % (Fig. 5) seems to be log-normal. If this is the case, the statistical results (mean value and semivariogram parameters) can be biased. The authors are invited to check data distribution and, in case, to make a logarithmic transformation.

- The authors, following Afzal et al. (2011), apply kriging in order to make a 3D interpolation of Cu content. It is not clear if authors use kriging or block kriging. The last procedure in particular (but even the first one) introduces a bias because the fractal behaviour refers to interpolated concentration and not to original data and this aspect may influence fractal analysis. I suggest adding comments on the consequences of the application of an interpolation method on the found fractal ranges.

- The paper basically presents a comparison between two methods of analysis, for this reason, more comments should be added in the conclusions instead of simply describing the results.

- The lines 268-274 refer to particular samples that could validate results, but the outcome is not clear.

- Many of the articles listed in References are not cited in the text.

---

## Author Comment (AC1) · 20 May 2019

1. We check the Cu data distribution of Pulang deposit. And the distribution of Cu data is log-normal. So we make a logarithmic transformation for the original data (Fig. 1). We revise the statistical results.The experimental semi–variogram of Cu data of Pulang deposit indicates a range and nugget effect of 320.0 m and 0.25, seperately(Fig. 2).

2. The 3D model of the distribution of Cu in the Pulang porphyry copper deposit was generated with ordinary kriging using the Datamine software. Fundamentally, the accuracy of the interpolation results mainly depends on whether the interpolation model could well fit the spatial distribution characteristics of the deposit. The original drillhole data of ore element concentrations were interpolated by using the ordinary kriging method to calculate the $V(\leq v)$ and $V(\geq v)$ enclosed by a concentration contour in a 3D model in this study. The method estimates values in un-sampled locations based on moving average of the variable of interest satisfying different dispersion forms of data. It is a spatial estimation method that provides a minimum error-variance estimate of any unsampled value. The correct variogram in kriging interpolation can guarantee the accuracy of the interpolation results. The accuracy of the spatial interpolation analysis is verified by comparing the difference between the measured values and the predicted values, so as to select the best variogram model. In order to test the variogram model, the cross-validation method was used to determine whether the parameters of the variogram model are correct(Fig. 3). The distribution of the residual is normal and the mean of error between the actual and estimated Cu grade values is equal to 0. It indicates that this model is reasonable, and the variogram parameters are unbiased for estimating the Cu grade.

3. In the many cases, drillcore logging in the field is dealing with the lack of proper diagnosis of geological phenomenon and it can undermine delineation of mineralized zones because it depends on the interpretation of individual loggers, which is subjective and no two loggers usually have the same interpretations. However, conventional geological modeling based on drillcore data is fundamentally important for ore body spatial structure understanding and mathematical applications. Grades of the ore elements are not observed in conventional methods of geological ore modeling while the variations in ore grades in a mineral deposit is an obvious and salient feature. Given the problems as mentioned above, using a series of newly established methods based on mathematical analyses such as fractal modeling seems to be inevitable. This study utilized the concentration–volume (C–V) and power spectrum–volume (S–V) fractal models to delineate and recognize different grade Cu mineralization of Pulang copper deposit. Both the fractal models reveal high grade Cu mineralization is located at the central and southern parts of Pulang deposit. The Cu threshold of high grade mineralization is 1.88% according to C–V method. And Cu threshold of supergene enrichment zones is 1.33% on the basis of S–V method. Models of moderate grade mineralization zones contain 1.38–1.88% Cu according to the C–V method. And the hypogene zones contain 0.23–1.33% Cu according to the S–V model. The C–V method shows barren host rocks include <0.25% and weak grade mineralization include 0.25–1.38% Cu. And the S–V model reveals that barren host rock and leached zone contain <0.23% Cu. Carranza (2011) has illustrated an analysis for calculation of spatial correlations between two binary especially mathematical and geological models. An intersection operation between the mineralization zones obtained from fractal models and different alteration zones in the geological model was performed to derive the amount of voxels corresponding to each of the classes of overlap zones. Using the obtained numbers of voxels, Type I error (T1E), Type II error (T2E), and overall accuracy (OA) of the fractal model were estimated with respect to different alteration zones due to geological data. And the values of OA of fractal models of mineralized zones were compared with each other. The comparison between highly mineralized zones on the basis of the fractal models and potassic zones resulted from 3D geological model illustrates that the S–V fractal model is better than the C–V model because the fact that the number of overlapped voxels (A) in the S–V model is higher than those in the C–V model. The overall accuracy values of C–V and S–V fractal models with respect to the potassic alteration zones of the geological model are 0.50 and 0.52, which illustrate that the S–V model gives better results to recognize high grade mineralization in Pulang deposit. On the other hand, correlation (from OA results) between highly mineralized zones obtained from S–V modeling and the potassic alteration zones is higher than the C–V model because of a strong proportional relationship between extension and positions of voxels in the S–V model and potassic alteration zones in the 3D geological model. Comparison between phyllic alteration zones resulted from the 3D geological model and moderate grade mineralization obtained from fractal methods indicates that OA values of C–V and S–V fractal methods in regard to phyllic alteration zones of the geological model are 0.59 and 0.56, respectively. The OA values of moderate and weak grade mineralization zones obtained from C–V model is higher than the results obtained by S–V model. On the other hand, moderately mineralized zones defined by C–V modeling have overlap with the phyllic alteration zones in the 3D geological model. However, the outcomes of the C–V model are more accurate than those of the S–V model with respect to the phyllic alteration zones in the 3D geological model. According to the correlation between results driven by fractal modeling and geological logging from drill holes in the Pulang porphyry copper deposit, high grade mineralization zones generated by fractal models, especially the S–V model, have a better correlation with potassic alteration zones resulted from the 3D geological model than the C–V model. And moderately mineralized zones correlate with phyllic alteration zones in the central and southern parts of the Pulang deposit. There is a better relationship between moderately and weakly mineralized zones derived by the C–V model and the phyllic alteration zones according to the 3D geological model than the S–V model.

4. Several samples were collected from different drill holes in different grade mineralization zones of Pulang deposit to validate the results of fractal models. They were analyzed by microscopic identification and XRF (X-ray Fluorescence Spectrometer), as depicted in Figure 4. PL-B82 sample was collected from the drill hole situated in the high grade mineralization zones. There are high chalcopyrite content and some molybdenite (Fig.14a). PL-B62 sample was collected from the drill hole situated in the moderate grade mineralization zones. There are low chalcopyrite content and some pyrrhotite content in polished section (Fig.14b). PL-B74 sample was collected from the drill hole located at the weakly mineralized zones with lower chalcopyrite content and some pyrrhotite (Fig.14c and Fig.14d). Results obtained from mineralogy, microscopic identification and drillcore scanning via XRF of these samples indicates that Cu values are 1.80%,1.32% and 0.41% in PL-B82, PL-B62 and PL-B74, respectively.

5. I have revised my manuscript. Many of the articles listed in References have been added and cited in the text. And a new revision of this manuscript has been uploaded.

Please also note the supplement to this comment:
https://www.nonlin-processes-geophys-discuss.net/npg-2019-8/npg-2019-8-AC1-supplement.pdf
* * *
[Figure]

The histogram of the Cu raw (a) and logarithmic transformation (b) data.

**Fig. 1.** The histogram of the Cu raw (a) and logarithmic transformation (b) data.

The experimental semi–variogramof Cu data in Pulangdeposit.

**Fig. 2.** The experimental semi–variogram of Cu data in Pulang deposit.

The cross-validation results: (a) residual VS Cu grade;(b) the residual distribution histogram.

**Fig. 3.** The cross-validation results: (a) residual VS Cu grade;(b) the residual distribution histogram.

| Sample no. | Mineralized zones obtained by fractal models | Cu(%) |
|---|---|---|
| PL-B74 | Weakly mineralized zones | 0.41 |
| PL-B62 | Moderately mineralized zones | 1.32 |
| PL-B82 | Highly mineralized zones | 1.80 |

**Fig. 4.** Results of XRF analysis of samples collected from different mineralized zones in the Pulang porphyry copper deposit.

**Supplement:**

# Application of fractal models to delineate mineralized zones in

# the Pulang porphyry copper deposit, Yunnan, Southwest China

Xiaochen Wang[a], Qinglin Xia[a,b,*], Tongfei Li[a], Shuai Leng[a], Yanling Li[a],

Li Kang[a], Zhijun Chen[a], Lianrong Wu[c]

[a] Faculty of Earth Resources, China University of Geosciences, Wuhan 430074, China

[b] Collaborative Innovation Center for Exploration of Strategic Mineral Resources,

Wuhan 430074, China

[c] Yunnan Diqing Nonferrous Metal Co., Ltd., Shangri-La 674400, China

## Abstract

The purpose of this study is to delineate various mineralized zones and the barren host rocks based on the surface and subsurface lithogeochemical data using the concentration–volume (C–V) and power spectrum–volume (S–V) fractal models in the Pulang porphyry copper deposit, southwest China. Results obtained by the concentration–volume model depict four geochemical zones defined by Cu thresholds of 0.25%, 1.38% and 1.88%, which represent non-mineralized wall rocks (Cu<0.25%), weakly mineralized zones (0.25%−1.38%), moderately mineralized zones (1.38%−1.88%), and highly mineralized zones (Cu>1.88%). S–V model is used by performing 3D fast Fourier transformation on assay data in the frequency domain.

The S–V method reveals three mineralized zones characterized by Cu threshold values of 0.23% and 1.33%. The zones of <0.23% Cu represent barren host rocks and zones of 0.23%-1.33% Cu represent the hypogene zones and zones >1.33% Cu represent supergene enrichment zones. Both the multifractal models show that high grade mineralization is located at the center and southern parts of Pulang deposit. The results are compared with the alteration and mineralogical models resulted from the

3D geological model using the logratio matrix method. The results show that the S–V

model gives better results to identify highly mineralized zones in the deposit.

However, the results of C–V method for moderately and weakly grade mineralization zones are more accurate than the zones obtained from S–V method.

[revised manuscript text omitted]
 of error between the actual and estimated Cu grade values is equal to 0 (Table 1). It indicates that this model is reasonable, and the variogram parameters are unbiased for estimating the Cu grade.

The obtained block models were used as input to the fractal models. The Pulang deposit was modeled by 20m × 20m × 5m voxels and they were decided by the grid drilling dimensions and geometrical properties of the deposit (David, 1970). The Pulang deposit is totally modeled with 150,973 voxels. The terms of "highly", "moderately" and "weakly" have been used to classify the mineralized zones based on fractal modeling and accordance with the classification of in terms of ore grades in the deposit.

<Fig. 5 inserts here>

<Fig. 6 inserts here>

**4.1. Concentration–volume (C–V) fractal modeling**

The occupied volume values corresponding to Cu grades were computed to obtain the concentration–volume model according to the 3D model of Pulang deposit. Through the obtained C–V log–log plot, the threshold values of Cu grades were determined (Fig.8). It indicates the power-law relationship between Cu grades and volumes. Three thresholds and four populations were obtained from C–V log–log plot, consequently. The first Cu threshold is 0.25%. The range of Cu values of <0.25% represent barren host rocks. The second Cu threshold is 1.38%, and values of 0.25–1.38% Cu represent weakly grade mineralization zones. And the third Cu threshold is 1.88%. The range of Cu values of 1.38–1.88% denote moderately mineralized zones, and values of >1.88% Cu indicate highly mineralized zones (Table 2). According to the results, the low concentration zones exist in many parts of Pulang deposit and are disposed along the northwest–southeast trend of the deposit. Moderately and highly mineralized zones are located at several parts of the center and south of Pulang deposit (Fig. 9).

**4.2. Power spectrum–volume (S–V) fractal modeling**

According to the geological data, such as the collar coordinates, azimuth, dip, mineralogy and lithology recorded from 130 drill holes, a 3D model and block model of Cu distribution of Pulang deposit were constructed with ordinary kriging method using the Geovia Surpac software.

The power spectrum (S) were computed for the 3D elemental distribution utilizing 3D fast Fourier transformation by MATLAB (R2016a). The logarithmic values of power spectrums and relevant volume values were plotted against each other (Fig. 10). The straight lines fitted through the log–log plot indicate different relationships between power spectrums and occupied volumes. The results have indicated that there are two thresholds and three different power–law relationships. The thresholds of logS=7.81 and logS=8.70 were decided by the log–log S–V plot. The 3D filters were designed to separate different mineralization zones on the basis of these threshold values. Inverse fast Fourier transformation was used to convert the decomposed components back into the space domain by MATLAB (R2016a). According to the results, Cu concentrations of the hypogene zones range from 0.23% to 1.33% (Table 3), and values of >1.33% Cu refer to the supergene enrichment zones, whereas values of <0.23% Cu pertain to the leached zone and barren host rocks (Fig. 11).

<Fig. 9 inserts here>

<Fig. 10 inserts here>

< Table 2 inserts here>

**5. Comparison of fractal models and geological model of the deposit**

Alteration models have a key role in zone delineation and also in presenting geological models, as described by Lowell and Guilbert (1970). The potassic and phyllic alterations control major mineralization within supergene enrichment and hypogene zones according to these models. The models of various mineralization zones obtained by the fractal methods could be compared with geological data to validate these results.

Results of fractal models of Pulang deposit were in contrast with the 3D

geological model of Pulang deposit constructed by utilizing Geovia Surpac software and drillholes data (Fig. 2). Furthermore, the results obtained from fractal models are also controlled by mineralogical investigations.

Carranza (2011) has illustrated an analysis for calculation of spatial correlations between two binary especially mathematical and geological models. An intersection operation between the mineralization zones obtained from fractal models and different alteration zones in the geological model was performed to derive the amount of voxels corresponding to each of the classes of overlap zones (Table 4). Using the obtained numbers of voxels, Type I error (T1E), Type II error (T2E), and overall accuracy (OA) of the fractal model were estimated with respect to different alteration zones due to geological data (Carranza, 2011). And the values of OA of fractal models of mineralized zones were compared with each other as follows.

The comparison between highly mineralized zones on the basis of the fractal models and potassic alteration zones resulted from the 3D geological model illustrates that the results of these two fractal models are similar. The overall accuracy values of

C–V and S–V models are 0.50 and 0.52 as shown in Table 5, which illustrate that the

S–V model gives more accurate results to recognize highly grade mineralization zones in Pulang deposit.

Comparison between phyllic alteration zones resulted from the 3D geological model and moderately grade mineralization zones obtained from fractal methods indicates that OA values of C–V and S–V fractal methods in regard to phyllic alteration zones of the geological model are 0.59 and 0.56 (Table 6). The OA values of moderately and weakly grade mineralization zones obtained from C–V model is higher than the results obtained from S–V model.

It could be considered that there are spatial correlations between different modeled Cu zones and geological features such as alterations and mineralogy. Several samples were collected from different drill holes in different grade mineralization zones of Pulang deposit to validate the results of fractal models. They were analyzed by microscopic identification and XRF (X-ray Fluorescence Spectrometer). PL-B82

sample was collected from the drill hole situated in the high grade mineralization zones. There are high chalcopyrite content and some molybdenite (Fig.14a). PL-B62

sample was collected from the drill hole situated in the moderate grade mineralization zones. There are low chalcopyrite content and some pyrrhotite content in polished section (Fig.14b). PL-B74 sample was collected from the drill hole located at the weakly mineralized zones with lower chalcopyrite content and some pyrrhotite (Fig.14c and Fig.14d). Results obtained from mineralogy, microscopic identification and drillcore scanning by XRF of these samples indicates that Cu concentrations are

1.80%,1.32% and 0.41% in PL-B82, PL-B62 and PL-B74 samples, respectively (Table 7).

**6. Conclusions**

In the many cases, drillcore logging in the field is dealing with the lack of proper diagnosis of geological phenomenon and it can undermine delineation of mineralized zones because it depends on the interpretation of individual loggers, which is subjective and no two loggers usually have the same interpretations. However, conventional geological modeling based on drillcore data is fundamentally important for ore body spatial structure understanding and mathematical applications. Grades of the ore elements are not observed in conventional methods of geological ore modeling while the variations in ore grades in a mineral deposit is an obvious and salient feature.

Given the problems as mentioned above, using a series of newly established methods based on mathematical analyses such as fractal modeling seems to be inevitable.

This study utilized the concentration–volume (C–V) and power spectrum–volume (S–V) fractal models to delineate and recognize different grade Cu mineralization zones of Pulang copper deposit. Both the fractal models reveal high grade Cu mineralization zones is located at the central and southern parts of Pulang deposit. The Cu threshold of high grade mineralization zones is 1.88% according to

C–V method. And Cu threshold of supergene enrichment zones is 1.33% on the basis of S–V method. Models of moderate grade mineralization zones contain 1.38–1.88%

Cu according to the C–V method. And the hypogene zones contain 0.23–1.33% Cu according to the S–V model. The C–V method shows barren host rocks include

<0.25% and weak grade mineralization include 0.25–1.38% Cu. And the S–V model reveals that barren host rock and leached zone contain <0.23% Cu.

Carranza (2011) has illustrated an analysis for calculation of spatial correlations between two binary especially mathematical and geological models. An intersection operation between the mineralization zones obtained from fractal models and different alteration zones in the geological model was performed to derive the amount of voxels corresponding to each of the classes of overlap zones. Using the obtained numbers of voxels, Type I error (T1E), Type II error (T2E), and overall accuracy (OA)

of the fractal models were estimated with respect to different alteration zones due to geological data. And the values of OA of fractal models of mineralized zones were compared with each other.

The comparison between highly mineralized zones based on the fractal models and potassic zones resulted from 3D geological model illustrates that the S–V fractal model is better than the C–V model because the fact that the number of overlapped voxels (A) in the S–V model is higher than those in the C–V model. The overall accuracy values of C–V and S–V fractal models with respect to the potassic alteration zones of the geological model are 0.50 and 0.52, which illustrate that the S–V model gives better results to recognize high grade mineralization zones in Pulang deposit.

On the other hand, correlation (from OA results) between highly mineralized zones obtained from S–V modeling and the potassic alteration zones is higher than the C–V

model because of a strong proportional relationship between extension and positions of voxels in the S–V model and potassic alteration zones in the 3D geological model.

Comparison between phyllic alteration zones resulted from the 3D geological model and moderate grade mineralization zones obtained from fractal methods indicates that OA values of C–V and S–V fractal methods in regard to phyllic alteration zones of the geological model are 0.59 and 0.56, respectively. The OA

values of moderate and weak grade mineralization zones obtained from C–V model is higher than the results obtained by S–V model. On the other hand, moderately mineralized zones defined by C–V modeling have overlap with the phyllic alteration zones in the 3D geological model. However, the outcomes of the C–V model are more accurate than those of the S–V model with respect to the phyllic alteration zones in the 3D geological model.

According to the correlation between results driven by fractal modeling and geological logging from drill holes in the Pulang porphyry copper deposit, high grade mineralization zones generated by fractal models, especially the S–V model, have a better correlation with potassic alteration zones resulted from the 3D geological model than the C–V model. And moderately mineralized zones correlate with phyllic alteration zones 
[revised manuscript text omitted]

[Figure]

**Fig. 1.**

[Figure]

[Figure]

              **Fig. 2.**

[Figure]

[Figure]

**Fig. 3.**

[Figure]

**Fig. 4.**

[Figure]

**704**

**705**

Fig. 5.

[Figure]

**706**

**707**

Fig. 6.

[Figure]

**708**
**709**
**710**
**711**
**712**
**713**

**714**

Fig. 7.

[Figure]

**Fig. 8.**

(a)

[Figure]

(b)

[Figure]

(c)

[Figure]

(d)

[Figure]

                                **Fig. 9.**

[Figure]

**Fig. 10.**
(a)

[Figure]

(b)

[Figure]

(c)

**Fig. 11.**

(a)

[Figure]

(b)

[Figure]

(c)

[Figure]

**Fig. 12.**

(a)

[Figure]

(b)

[Figure]

(c)

[Figure]

          **Fig. 13.**

[Figure]

**Fig. 14.**

**Table 1**

| Variables | Residual |
|---|---|
| Mean | 0.000 |
| Variance | 0.016 |
| Standard Deviation | 0.127 |

**Table 2**

| Mineralized zones | Thresholds(Cu%) | Range(Cu%) |
|---|---|---|
| Barren host rock | | <0.25 |
| Weakly mineralized | 0.25 | 0.25–1.38 |
| Moderately mineralized | 1.38 | 1.38–1.88 |
| Highly mineralized | 1.88 | >1.88 |

**Table 3**

| Mineralized zones | PS threshold | Range of PS | Range(Cu%) |
|---|---|---|---|
| leached zone and barren host rock | | <7.81 | <0.23 |
| hypogene zones | 7.81 | 7.81-8.70 | 0.23-1.33 |
| supergene enrichment zones | 8.70 | >8.70 | >1.33 |

**Table 4**

| | | Geological model | |
|---|---|---|---|
| | | Inside zone | Outside zone |
| Fractal model | Inside zone | True positive (A) | False positive (B) |
| | Outside zone | False negative (C) | True negative (D) |
| | | TypeIerror=C/(A+C) | TypeIIerror=B/(B+D) |
| | | Overallaccuracy=(A+D)/(A+B+C+D) | |

**Table 5**

|  |  | Potassic alteration of geological model | |
| --- | --- | --- | --- |
|  |  | Inside zones | Outside zones |
| C–V fractal model of highly mineralized zones | Inside zones | A   2850 | B       1360 |
|  | Outside zones | C   77927 | D      76913 |
|  |  | T1E   0.96 | T2E   0.02 |
|  |  | OA | 0.50 |
| S–V fractal model of supergene enrichment zones | Inside zones | A   4131 | B   2318 |
|  | Outside zones | C   73985 | D   74726 |
|  |  | T1E   0.95 | T2E   0.03 |
|  |  | OA | 0.52 |

**Table 6**

|  |  | Phyllic alteration of geological model | |
| --- | --- | --- | --- |
|  |  | Inside zones | Outside zones |
| C–V fractal model of moderately and weakly mineralized zones | Inside zones | A    36518 | B      48027 |
|  | Outside zones | C    25461 | D      69155 |
|  |  | T1E   0.41 | T2E   0.40 |
|  |  | OA | 0.59 |
| S–V fractal model of the hypogene zones | Inside zones | A   40080 | B   44943 |
|  | Outside zones | C   26899 | D   54239 |
|  |  | T1E   0.40 | T2E   0.45 |
|  |  | OA | 0.56 |

**Table 7**

| Sample no. | Mineralized zones obtained by fractal models | Cu(%) |
| --- | --- | --- |
| PL-B74 | Weakly mineralized zones | 0.41 |
| PL-B62 | Moderately mineralized zones | 1.32 |
| PL-B82 | Highly mineralized zones | 1.80 |

---

## Referee Comment (RC2) · Anonymous Referee #2 · 24 Jun 2019

Dear authors,

Regarding your submitted paper I must say, in general considering its results, it could be a nice paper, but it needs to be improved in some cases: 1. Typing/spacing issues 2. Grammatical issues 3. You can improve the paper with many other, even newer references. 4. In some cases, the paper is prolonged by repeating obvious things. For example, about the amounts of the thresholds, some tables could be informative enough and no need to mention them. 5. Honestly to me, there was nothing new in this paper and the paper was totally like what Afzal et al have done but on a different case study. This is acceptable, but the readers may need at least a very tiny interesting,

innovative or new thing in it. If you make something different, for example from a different point of view, it would make a big bonus for your paper.

So, I believe if you revolutionize the style of the paper or even add one more fractal model, like N-S fractal model, to improve and change the structure of the paper, it would be great.

So, considering all these points, my suggestion is: Major Revision. However, it is potentially acceptable after improvements.

Best,

---

## Author Comment (AC2) · 9 Jul 2019

1. We have checked this paper and the typing/spacing issues have been revised. 2. We have checked this paper and revised the grammatical issues. A new revision of this manuscript has been uploaded. 3. We have added many newer references to improve the paper. And a new revision of this manuscript has been uploaded. 4. We have checked this paper and found that the paper is prolonged by repeating obvious things for example the amounts of the thresholds. We have deleted these obvious things. And a new revision of this manuscript has been uploaded. 5. Given the Referee comments, we have tried to add the N-S fractal model to improve the structure of the paper and

revolutionize the style of the paper. Furthermore, the results of N-S fractal model were compared with the C-V and S-V models.

5.1 Number-size (N-S) fractal model

Number-size (N-S) method proposed by Mandelbrot (1983) can be utilized to describe the distribution of geochemical populations (Sadeghi et al., 2012). In this method, geochemical data do not undergo any pre-processing (Mao et al., 2004). This model shows a relationship between desirable attributes (e.g. Cu concentration in this study) and their cumulative number of samples (Sadeghi et al., 2012). A power-law frequency model has been proposed to explain the N-S relationship according to the frequency distribution of elemental concentrations and cumulative number of samples with those attributes (e.g., Li et al., 1994; Sadeghi et al., 2012; Sanderson et al., 1994; Shi and Wang, 1998; Turcotte, 1996; Zuo et al., 2009a).

The N-S model proposed by Mandelbrot (1983) has been expressed as follows: $N(\geq p)=Fp-D$

where p denotes element concentration, $N(\geq p)$ denotes cumulative number of samples with concentration values greater than or equal to p, F is a constant and D is the scaling exponent or fractal dimension of the distribution of element concentrations. According to Mandelbrot (1983), log-log plots of $N(\geq p)$ versus p show straight line segments with different slopes -D corresponding to different concentration intervals.

5.2 Number-size (N-S) fractal modeling

The N-S model was applied to the Cu data (Fig. 8). The selection of breakpoints as threshold values appears to be an objective decision because geochemical populations are defined by different line segments in the N-S log-log plot. The straight fitted lines were obtained based on least-square regression (Agterberg et al., 1996; Spalla et al., 2010). In other words, the intensity of element enrichment is depicted by each slope of the line segment in the N-S log-log plots (Afzal et al., 2010; Bai et al., 2010). Based on the classification of the 3D model of Cu data and the thresholds obtained from N-S fractal model (Table 2), highly mineralized zones are situated in the southern and central parts of Pulang deposit that coincide with the potassium-silicate alterations. However, small highly mineralized zones are located in the central parts of the Pulang deposit (Fig.9). Moderately mineralized zones are disposed in a northwest-southeast trend correlated with phyllic zones. Weakly mineralized zones and barren host rocks are situated in the marginal parts of the area.

A comparison between highly mineralized zones based on the fractal models and potassic alteration zones resulted from the 3D geological model shows that there is a similarity among these fractal models. Overall accuracies for the C-V, N-S and S-V models are 0.50, 0.51 and 0.52, respectively (Table 6), which indicate that the S-V model gives better results to identify highly mineralized zones in the deposit. Because the fact that the number of overlapped voxels (A) in the S-V model is higher than those in N-S and C-V model. The correlation (from OA results) between highly mineralized zones obtained from S-V modeling and the potassic alteration zones is better than the N-S and C-V model because of a strong proportional relationship between extension and positions of voxels in the S-V model and potassic alteration zones in the 3D geological model.

Comparison between phyllic alteration zones resulted from the 3D geological model and moderately and weakly mineralized zones from fractal modeling shows that overall accuracies of the C-V, N-S and S-V fractal models with respect to phyllic alteration zones of the geological model are 0.59, 0.56 and 0.54, respectively. Overall accuracy values of moderately and weakly mineralized zones obtained from C-V modeling is higher than the mineralized zones obtained from N-S and S-V modeling (Table 7). On the other hand, moderately mineralized zone defined by C-V modeling has overlap with the phyllic zones in the 3D geological model. However, the results of the C-V model are more accurate than those of the N-S and S-V model with respect to the phyllic zones in the 3D geological model.

And a new revision of this manuscript has been uploaded.

Please also note the supplement to this comment:
https://www.nonlin-processes-geophys-discuss.net/npg-2019-8/npg-2019-8-AC2-supplement.pdf

———————————————————

[Figure]

[Figure]

**Fig. 1.** N–S log–log plot for Cu concentrations in the Pulang deposit.

[Figure]

**Fig. 2.** Zones in Pulang deposit based on thresholds defined from N–S fractal model of Cu data: (a) highly mineralized zones; (b) moderately mineralized zones; (c) weakly mineralized zones and barren host rock

| Mineralized zones | Thresholds(Cu%) | Range(Cu%) |
|---|---|---|
| Barren host rock and weakly mineralized | | <0.28 |
| Moderatelymineralized | 0.28 | 0.28-1.45 |
| Highly mineralized | 1.45 | >1.45 |

**Fig. 3.** Thresholds concentrations obtained by using N-S model based on Cu% in Pulang deposit.

**Table 6**

| | | Potassic alteration of geological model | |
|---|---|---|---|
| | | Inside zones | Outside zones |
| C–V fractal model of highly mineralized zones | Inside zones | A 2850 | B 1360 |
| | Outside zones | C 77927 | D 76913 |
| | | T1E 0.96 | T2E 0.02 |
| | | OA | 0.50 |
| N–S fractal model of highly mineralized zones | Inside zones | A 3092 | B 1570 |
| | Outside zones | C 75025 | D 75473 |
| | | T1E 0.96 | T2E 0.02 |
| | | OA | 0.51 |
| S–V fractal model of supergene enrichment zones | Inside zones | A 4431 | B 2318 |
| | Outside zones | C 72985 | D 75726 |
| | | T1E 0.94 | T2E 0.03 |
| | | OA | 0.52 |

**Fig. 4.** Overall accuracy (OA), Type I and Type II errors with respect to potassic alteration zone resulted from geological model and threshold values of Cu obtained through C–V , N–S and S–V fractal modeling.

**Table 7**

| | | Phyllic alteration of geological model | | | |
|---|---|---|---|---|---|
| | | Inside zones | | Outside zones | |
| C–V fractal model of moderately and weakly mineralized zones | Inside zones | A | 36518 | B | 48027 |
| | Outside zones | C | 25461 | D | 69155 |
| | | T1E | 0.41 | T2E | 0.40 |
| | | OA | | 0.59 | |
| N–S fractal model of moderately mineralized zones | Inside zones | A | 40080 | B | 44943 |
| | Outside zones | C | 26899 | D | 54239 |
| | | T1E | 0.40 | T2E | 0.45 |
| | | OA | | 0.56 | |
| S–V fractal model of the hypogene zones | Inside zones | A | 35555 | B | 46943 |
| | Outside zones | C | 23955 | D | 48223 |
| | | T1E | 0.40 | T2E | 0.49 |
| | | OA | | 0.54 | |

**Fig. 5.** Overall accuracy (OA), Type I and Type II errors with respect to phyllic alteration zone resulted from geological model and threshold values of Cu obtained through C–V, N–S and S–V fractal modeling.

**Supplement:**

**Application of fractal models to delineate mineralized zones in the Pulang porphyry copper deposit,Yunnan, Southwest China**

Xiaochen Wang[a], Qinglin Xia[a,b,*], Tongfei Li[a], Shuai Leng[a], Yanling Li[a], Li Kang[a], Zhijun Chen[a], Lianrong Wu[c]

[a] Faculty of Earth Resources, China University of Geosciences, Wuhan 430074, China

[b] Collaborative Innovation Center for Exploration of Strategic Mineral Resources, Wuhan 430074, China

[c] Yunnan Diqing Nonferrous MetalCo., Ltd., Shangri-La 674400, China

**Abstract**

The aim of this study is to delineate and recognize various mineralized zones and barren host rocks based on the surface and subsurface lithogeochemical data utilizing the number-size (N-S), concentration-volume (C-V) and power spectrum-volume (S-V) fractal models in the Pulang porphyry copper deposit, southwest China. The N-S model reveals three mineralized zones characterized by Cu thresholds of 0.28% and 1.45%, with zones <0.28% Cu representing weakly mineralized zones and barren host rocks, with zones 0.28%-1.45% Cu representing moderately mineralized zones and zones >1.45% Cu representing highly mineralized zones. Results obtained by the C-V model depict four geochemical zones defined by Cu thresholds of 0.25%, 1.48% and 1.88%, which represent non-mineralized wall rocks (Cu<0.25%), weakly mineralized zones (0.25%-1.48%), moderately mineralized zones (1.48%-1.88%), and highly mineralized zones (Cu>1.88%). S-V model is used by performing 3D fast Fourier transformation on assay data in the frequency domain. The S-V model reveals three mineralized zones characterized by Cu thresholds of 0.23% and 1.33%, with zones of <0.23% Cu representing leached zone and barren host rocks, with zones of 0.23%-1.33% Cu representing the hypogene zones and zones of >1.33% Cu representing 
[revised manuscript text omitted]

$\mu(x, y, z)$; Wx, Wy and Wz respectively indicate wave numbers or angular frequencies in X, Y and Z axes directions on a 3D model. The range of index $\beta$ is $0<\beta \leq 2$ or $1 \leq 2/\beta$

with the special case of β=2 or 2/β=1 corresponding to non-fractal or monofractal and

1<2/β to multifractals (Cheng, 2006).

By using the method of geostatistical estimation, the drillhole data of elemental concentration values were interpolated to construct the block model with ore element distribution. The power spectrum values can be obtained by using 3D fast Fourier transformation for ore element grades. The logarithm of all power spectrum values and accumulative volume values were calculated. And the log-log plot between power spectrums and volumes was drawn according to previous counted values. Then the filters were constructed on the basis of threshold values obtained by the log-log plot of

S-V. Finally, the power spectrums were converted back to the space domain by utilizing inverse fast Fourier transformation.

**3. Geological setting of the Pulang porphyry copper deposit**

The Pulang porphyry copper deposit is situated in the southern end of the Yidun continental arc, southwest China (Fig.1). The continental arc was produced due to the westward subduction of Garze–Litang oceanic crust (Deng et al., 2014b, 2015; Wang et al., 2014). And the Pulang ore deposit, one of the largest porphyry copper deposits in

China (Deng et al., 2012, 2014a; Mao et al., 2012, 2014), is characterized by typical porphyry-type alteration zone. The geological characteristics of the deposit, including the alteration types and their zonation, the geometry of orebody, metallogenic time and the geodynamic settings have been systematically researched (Leng et al., 2012; Li et al.,

2011, 2013). The deposit consists of five ore-bearing porphyry bodies, covering an area of approximately 9 km$^2$ , and the explored ore tonnage of Cu is estimated to be 6.50 Mt (Liu et al., 2013).

The outcrop strata of Pulang deposit are dominated by Upper Triassic Tumugou

Formation clastic rocks and andesite, and Quaternary sediments (Fig.1c). The Triassic porphyry intrusions primarily comprise quartz diorite porphyry, quartz monzonite porphyry, quartz diorite porphyrite and granodiorite porphyry.The Tumugou Formation strata was intruded by the quartz diorite porphyry with an age of 219.6 ± 3.5 Ma (Zircon U-Pb dating) (Pang et al., 2009). Then quartz monzonite porphyry with an age of 212.8 ± 1.9 Ma and granodiorite porphyry with an age of 206.3 ± 0.7 Ma (Zircon

U-Pb dating) (Liu et al., 2013) crosscut quartz diorite porphyry, respectively. The quartz monzonite porphyry is considered to be associated with mineralization because its age is similar with the molybdenite Re-Os isochron age of 213 ± 3.8 Ma from orebody (Zeng et al., 2004). Moreover, the Cu concentrations of quartz monzonite porphyry are higher than the other porphyries.

The porphyry-type alteration zones transit upward and outward from early potassium-silicate, through quartz-sericite to propylitization from the core of the quartz monzonite porphyry (Fig. 4). The wall rocks near the porphyries were mostly changed into hornfels. Systematic drilling has demonstrated that the potassium-silicate and quartz-sericite zones host the main orebodies, constituting the core of mineralized zones. And the propylitic zones and hornfels only develop the weak mineralization.

The orebodies occur mainly in potassium-silicate and quartz-sericite, and occur as veins in the propylitic zones and hornfels. Major rock types in the deposit are quartz monzonite porphyry, quartz diorite porphyrite, granite diorite porphyry, quartz diorite porphyry and hornfels (Fig.2). Metallic minerals mainly include pyrite, chalcopyrite with small amount of molybdenite and pyrrhotite (Fig. 3).

**4. Fractal modeling**

Based on the geological data (which include collar coordinates of each drillhole, azimuth and dip (orientation), lithology and mineralogy) recorded from 130 drillholes in Pulang deposit, 20492 lithogeochemical samples have been collected at 2 m intervals. The laboratory of the 3rd Geological Team of the Yunnan Bureau of Geology and Mineral Resources utilized the iodine-fluorine and oscillo-polarographic method to analyze the concentrations of Cu and associated paragenetic elements and its analytical uncertainty is less than 7% (Yunnan Diqing Nonferrous Metal Co. Ltd., 2009). Only

Cu concentrations were researched in this study. The distribution of Cu concentrations is log-normal (Fig. 5). The experimental semi-variogram of Cu data of Pulang deposit indicates a range and nugget effect of 320.0m and 0.25, seperately (Fig. 6). The spherical model is fitted in regard to the experimental semi-variogram. The 3D model of Cu concentrations distribution of Pulang deposit was produced with ordinary kriging method using the Geovia Surpac software on the basis of the semi-variogram and anisotropic ellipsoid. Fundamentally, the accuracy of the interpolation results mainly depends on whether the interpolation model could well fit the spatial distribution characteristics of the deposit. Ordinary kriging was used because it is compatible with a stationary model; it only involves a variogram, and it is in fact the form of kriging used most (Chilès and Delfiner, 1999). Goovaerts (1997) showed that the values in un-sampled locations are estimated by the ordinary kriging method according to moving average of the interest variables satisfying various distribution forms of data. It is a spatial estimation method where the error variance is minimized. This error variance is based on the configuration of the data and its variogram (Yamamoto, 2005). The correct variogram in kriging interpolation can guarantee the accuracy of the interpolation results.

The accuracy of the spatial interpolation analysis is verified by comparing the difference between the measured values and the predicted values, so as to select the best variogram model. In order to test the variogram model, the cross-validation method was used to determine whether the parameters of the variogram model are correct. The distribution of the residual is normal (Fig.7) and the mean of error between the actual and estimated Cu grade values is equal to 0 (Table 1). It indicates that this model is reasonable, and the variogram parameters are unbiased for estimating the Cu grade.

The obtained block models were used as input to the fractal models. The Pulang deposit was modeled by 20m×20m×5m voxels and they were decided by the grid drilling dimensions and geometrical properties of the deposit (David, 1970). The Pulang deposit is totally modeled with 150,973 voxels. The terms of "highly", "moderately" and "weakly" have been used to classify the mineralized zones based on fractal modeling and accordance with the classification of in terms of ore grades in the deposit.

**4.1 Number-size (N-S) fractal modeling**

The N-S model was applied to the Cu data (Fig. 8). The selection of breakpoints as threshold values appears to be an objective decision because geochemical populations are defined by different line segments in the N-S log-log plot. The straight fitted lines were obtained based on least-square regression (Agterberg et al., 1996; Spalla et al.,2010). In other words, the intensity of element enrichment is depicted by each slope of the line segment in the N-S log-log plots (Afzal et al., 2010; Bai et al., 2010).

Based on the classification of the 3D model of Cu data and the thresholds obtained from N-S fractal model (Table 2), highly mineralized zones are situated in the southern and central parts of Pulang deposit that coincide with the potassium-silicate alterations.

However, small highly mineralized zones are located in the central parts of the Pulang deposit (Fig.9). Moderately mineralized zones are disposed in a northwest-southeast trend correlated with phyllic zones. Weakly mineralized zones and barren host rocks are situated in the marginal parts of the area.

**4.2. Concentration-volume (C-V) fractal modeling**

The occupied volume values corresponding to Cu grades were computed to obtain the concentration-volume model according to the 3D model of Pulang deposit.

Through the obtained C-V log-log plot, the threshold values of Cu grades were determined (Fig.10). It indicates the power-law relationships between Cu grades and volumes. According to these results (Table 3), the low concentration zones exist in many parts of the deposit and are disposed along the NW-SE trend. Moderately and highly mineralized zones are situated in several parts of the center and south of the deposit (Fig. 11).

**4.3. Power spectrum-volume (S-V) fractal modeling**

Based on the geological data (which include collar coordinates of each drillhole, azimuth and dip (orientation), lithology and mineralogy) recorded from 130 drillholes in the deposit, a 3D model and block model of the distribution of Cu in Pulang deposit were constructed with ordinary kriging using the Geovia Surpac software.

The power spectrum (S) were calculated for the 3D elemental distribution using

3D fast Fourier transformation by MATLAB (R2016a). The logarithmic values of power spectrums and relevant volume values were plotted against each other (Fig. 12).

The straight lines fitted through log-log plot indicate different relationships between power spectrums and occupied volumes. The thresholds of logS=7.81 and logS=8.70

were decided by the log-log S-V plot. The 3D filters were designed to separate different mineralization zones on the basis of these threshold values. Inverse fast

Fourier transformation was used to convert the decomposed components back into the space domain by MATLAB (R2016a). According to the results, Cu concentrations of the hypogene zones range from 0.23% to 1.33% (Table 4), and values of >1.33% Cu refer to the supergene enrichment zones, whereas values of <0.23% Cu pertain to the leached zone and barren host rocks (Fig. 13).

**271  5. Comparison of fractal models and geological model of the deposit**

Alteration models have a key role in zone delineation and also in presenting geological models, as described by Lowell and Guilbert (1970). The potassic and phyllic alterations control major mineralization within supergene enrichment and hypogene zones according to these models. Models of Cu mineralization zones derived via the fractal models can be compared with geological data in order to validate the results of analysis in different porphyry Cu deposits. Results of fractal modeling of

Pulang deposit were compared with the 3D geological model of the deposit constructed by using the Geovia Surpac software and drillhole data (Fig. 2). Moreover, the results obtained from these fractal models are controlled by mineralogical investigations.

Carranza (2011) has illustrated an analysis for calculation of spatial correlations between two binary especially mathematical and geological models. An intersection operation between the mineralization zones obtained from fractal models and different alteration zones in the geological model was performed to derive the amount of voxels corresponding to each of the classes of overlap zones (Table 5). Using the obtained numbers of voxels, Type I error (T1E), Type II error (T2E), and overall accuracy (OA)

of the fractal model were estimated with respect to different alteration zones due to geological data (Carranza, 2011). The values of OA of fractal models of mineralized zones were compared with each other as follows.

A comparison between highly mineralized zones based on the fractal models and potassic alteration zones resulted from the 3D geological model shows that there is a similarity among these fractal models. Overall accuracies for the C-V, N-S and S-V

models are 0.50, 0.51 and 0.52, respectively (Table 6), which indicate that the S-V

model gives better results to identify highly mineralized zones in the deposit. Because the fact that the number of overlapped voxels (A) in the S-V model is higher than those in N-S and C-V model. The correlation (from OA results) between highly mineralized zones obtained from S-V modeling and the potassic alteration zones is better than the

N-S and C-V model because of a strong proportional relationship between extension and positions of voxels in the S-V model and potassic alteration zones in the 3D

geological model.

Comparison between phyllic alteration zones resulted from the 3D geological model and moderately and weakly mineralized zones from fractal modeling shows that overall accuracies of the C-V, N-S and S-V fractal models with respect to phyllic alteration zones of the geological model are 0.59, 0.56 and 0.54, respectively. Overall accuracy value of moderately and weakly mineralized zones obtained from C-V

modeling is higher than the mineralized zones obtained from N-S and S-V modeling (Table 7). On the other hand, moderately mineralized zones defined by C-V modeling have overlap with the phyllic zones in the 3D geological model. However, the results of the C-V model are more accurate than those of the N-S and S-V model with respect to the phyllic zones in the 3D geological model.

It could be considered that there are spatial correlations between different modeled Cu zones and geological features such as alterations and mineralogy. Several samples were collected from different drillholes in different grade mineralization zones of Pulang deposit to validate the results of fractal models. They were analyzed by microscopic identification and XRF (X-ray Fluorescence Spectrometer). The PL-B82

sample was collected from the drillhole situated in the high grade mineralization zones.

There are high chalcopyrite content and some molybdenite (Fig.16a). PL-B62 sample was collected from the drillhole situated in the moderate grade mineralization zones.

There are low chalcopyrite content and some pyrrhotite content in polished section (Fig.16b). PL-B74 sample was collected from the drillhole located at the weakly
mineralized zones with lower chalcopyrite content and some pyrrhotite (Fig.16c and
Fig.16d). Results obtained from mineralogy, microscopic identification and drillhole
scanning by XRF of these samples indicates that Cu concentrations are 1.80%, 1.32%
and 0.41% in PL-B82, PL-B62 and PL-B74 samples, respectively (Table 8).

**6. Conclusions**

In the many cases, drillhole logging is dealing with the lack of proper diagnosis of
geological phenomenon and it can undermine delineation of mineralized zones because
it depends on the interpretation of individual loggers, which is subjective and no two
loggers have the same interpretations. However, the conventional geological modeling
based on drillhole data is fundamentally important for ore body spatial structure
understanding and mathematical applications. Grades of the ore elements are not
observed in conventional methods of geological ore modeling while the variations in
ore grades in a mineral deposit is an obvious and salient feature. Given the problems as
mentioned above, using a series of new established methods based on mathematical
analyses such as fractal modeling seems to be inevitable.

In this paper, the number-size (N-S), concentration-volume (C-V) and power
spectrum-volume (S-V) fractal models were used to delineate and recognize various
Cu mineralized zones of Pulang porphyry copper deposit in the south end of the Yidun
continental arc, SW China. All the fractal models reveal high grade Cu mineralized
zones are situated in the central and southern parts of the deposit. The Cu threshold
values of highly mineralized zones are 1.45% and 1.88% based on the N-S and C-V
fractal models. And the Cu threshold of supergene enrichment zones is 1.33% based on
the S-V fractal model. Models of moderately mineralized zones contain 0.28-1.45% Cu
according to the N-S model, and 1.48-1.88% Cu according to the C-V model. The
hypogene zones contain 0.23-1.33% Cu according to the S-V model. The N-S model
reveals weakly mineralized zones and barren host rocks containing <0.28% Cu. In
contrast, the C-V model reveals that barren host rocks contain <0.25% and weakly
mineralized zones contain 0.25-1.48% Cu. And the S-V model reveals that barren host rock and leached zone contain <0.23% Cu.

The comparison between highly mineralized zones based on the fractal models and potassic zones resulted from 3D geological model illustrates that the S-V fractal model is better than the N-S and C-V model because the fact that the number of overlapped voxels (A) in the S-V model is higher than those in the N-S and C-V

model. Overall accuracies for the C-V, N-S and S-V models are 0.50, 0.51 and 0.52, respectively (Table 6), which indicate that the S-V model gives the best results to identify highly mineralized zones in the deposit. On the other hand, the correlation (from OA results) between highly mineralized zones obtained from S-V modeling and the potassic alteration zones is better than the N-S and C-V model because of a strong proportional relationship between extension and positions of voxels in the S-V model and potassic alteration zones in the 3D geological model.

Comparison between phyllic alteration zones resulted from the 3D geological model and moderate grade mineralization zones obtained from fractal models indicates that OA values of C-V, N-S and S-V fractal methods in regard to phyllic alteration zones of the geological model are 0.59, 0.56 and 0.54, respectively. Overall accuracy of moderately and weakly mineralized zones obtained from C-V modeling is higher than the mineralized zones obtained from N-S and S-V modeling (Table 7).

According to the correlation between the results driven by fractal modeling and geological logging from drillholes in the Pulang porphyry copper deposit, high grade mineralization zones generated by fractal models, especially the S-V model, has a better correlation with potassic alteration zones resulted from the 3D geological model than N-S and C-V model. 
[revised manuscript text omitted]

[Figure]

                    **Fig. 1.**

[Figure]

[Figure]

**Fig. 2.**

[Figure]

[Figure]

**Fig. 3.**

[Figure]

**Fig. 4.**

[Figure]

**Fig. 5.**

**Fig. 6.**

**Fig. 7.**

[Figure]

**Fig. 8.**

(a)

[Figure]

(b)

[Figure]

(c)

[Figure]

              **Fig. 9**

[Figure]

              **Fig. 10.**

(a)

[Figure]

(b)

[Figure]

(c)

[Figure]

(d)

[Figure]

**Fig. 11.**

[Figure]

                    **Fig. 12.**

(a)

[Figure]

(b)

[Figure]

(c)

[Figure]

                    **Fig. 13.**

(a)

[Figure]

(b)

[Figure]

(c)

[Figure]

(d)

[Figure]

**Fig. 14.**

(a)

[Figure]

(b)

[Figure]

(c)

[Figure]

(d)

[Figure]

**Fig. 15.**

[Figure]

**Fig. 16.**

**Table 1**

| Variables | Residual |
|---|---|
| Mean | 0.000 |
| Variance | 0.016 |
| Standard Deviation | 0.127 |

**Table 2**

| Mineralized zones | Thresholds(Cu%) | Range(Cu%) |
|---|---|---|
| Barren host rock and weakly mineralized | | <0.28 |
| Moderatelymineralized | 0.28 | 0.28-1.45 |
| Highly mineralized | 1.45 | >1.45 |

**Table 3**

| Mineralized zones | Thresholds(Cu%) | Range(Cu%) |
|---|---|---|
| Barren host rock | | <0.25 |
| Weakly mineralized | 0.25 | 0.25–1.48 |
| Moderately mineralized | 1.48 | 1.48–1.88 |
| Highly mineralized | 1.88 | >1.88 |

**Table 4**

| Mineralized zones | PS threshold | Range of PS | Range(Cu%) |
|---|---|---|---|
| leached zone and barren host rock | | <7.81 | <0.23 |
| hypogene zones | 7.81 | 7.81-8.70 | 0.23-1.33 |
| supergene enrichment zones | 8.70 | >8.70 | >1.33 |

**Table 5**

| | | Geologicalmodel Inside zone | Outside zone |
|---|---|---|---|
| Fractal model | Inside zone Outside zone | True positive (A) False negative (C) TypeIerror=C/(A+C) Overallaccuracy=(A+D)/(A+B +C+D) | False positive (B) True negative (D) TypeIIerror=B/(B+D) |

**Table 6**

| | | Potassic alteration of geological model | |
| --- | --- | --- | --- |
| | | Inside zones | Outside zones |
| C–V fractal model of highly mineralized zones | Inside zones | A 2850 | B 1360 |
| | Outside zones | C 77927 | D 76913 |
| | | T1E 0.96 | T2E 0.02 |
| | | OA | 0.50 |
| N–S fractal model of highly mineralized zones | Inside zones | A 3092 | B 1570 |
| | Outside zones | C 75025 | D 75473 |
| | | T1E 0.96 | T2E 0.02 |
| | | OA | 0.51 |
| S–V fractal model of supergene enrichment zones | Inside zones | A 4431 | B 2318 |
| | Outside zones | C 72985 | D 75726 |
| | | T1E 0.94 | T2E 0.03 |
| | | OA | 0.52 |

**Table 7**

| | | Phyllic alteration of geological model | |
| --- | --- | --- | --- |
| | | Inside zones | Outside zones |
| C–V fractal model of moderately and weakly mineralized zones | Inside zones | A 36518 | B 48027 |
| | Outside zones | C 25461 | D 69155 |
| | | T1E 0.41 | T2E 0.40 |
| | | OA | 0.59 |
| N–S fractal model of moderately mineralized zones | Inside zones | A 40080 | B 44943 |
| | Outside zones | C 26899 | D 54239 |
| | | T1E 0.40 | T2E 0.45 |
| | | OA | 0.56 |
| S–V fractal model of the hypogene zones | Inside zones | A 35555 | B 46943 |
| | Outside zones | C 23955 | D 48223 |
| | | T1E 0.40 | T2E 0.49 |
| | | OA | 0.54 |

**Table 8**

| Sample no. | Mineralized zones obtained by fractal models | Cu(%) |
| --- | --- | --- |
| PL-B74 | Weakly mineralized zones | 0.41 |
| PL-B62 | Moderately mineralized zones | 1.32 |
| PL-B82 | Highly mineralized zones | 1.80 |

---

## Author Response (AR1)

We would like to thank the editor and reviewers for giving us insightful suggestions which would help us in depth to improve the quality of the paper. We made a significant revision and the detailed responses are as follows.

Response to Referee comment 1:

Specific comments:

The language is quite poor as it presents some traduction and grammar errors and it is sometimes difficult to follow the logic of the text. Some parts are rather obscure (e.g. lines 123-124 or 249-253).

**Response:** We have tried our best to improve the grammatical errors and also consulted an English speaker. We hope it will meet with approval. A revision by a mother-tongue has been uploaded. The lines 123-124 have been revised as the lines 149-157 in a new revision of this manuscript. The lines 249-253 have been revised as the lines 286-294 in a new revision of this manuscript.

1. The histogram of Cu % (Fig. 5) seems to be log-normal. If this is the case, the statistical results (mean value and semivariogram parameters) can be biased. The authors are invited to check data distribution and, in case, to make a logarithmic transformation.

**Response 1:** Thank you. Accept this point. We have checked the Cu data distribution of Pulang deposit. We made a logarithmic transformation for the original data. The histogram and Q-Q plot of the log-transformed Cu data indicate that the distribution of Cu data is log-normal (Fig. 5). We revised the statistical results. The experimental semivariogram of Cu data of Pulang deposit indicates a range and nugget effect of 320.0 m and 0.25, respectively (Fig. 6).

**Fig. 5.** Histograms of (a) the Cu raw and (b) logarithmic transformation data and (c) Q-Q plot of the log-transformed Cu data in the Pulang deposit.

Fig. 6. The experimental semivariogram of Cu data in Pulang deposit.

2. The authors, following Afzal et al. (2011), apply kriging in order to make a 3D interpolation of Cu content. It is not clear if authors use kriging or block kriging. The last procedure in particular (but even the first one) introduces a bias because the fractal behaviour refers to interpolated concentration and not to original data and this aspect may influence fractal analysis. I suggest adding comments on the consequences of the application of an interpolation method on the found fractal ranges.

**Response 2**: Thank you. Accept this point. The 3D model of the Cu concentration distribution of the Pulang deposit was produced with the ordinary kriging method using Geovia Surpac software on the basis of the semivariogram and anisotropic ellipsoid. Fundamentally, the accuracy of the interpolation results mainly depends on whether the interpolation model accurately fits the spatial distribution characteristics of the deposit. The original drillhole data of ore element concentrations were interpolated by using the ordinary kriging method to calculate the  $V(\leq v)$  and  $V(\geq v)$  enclosed by a concentration contour in a 3D model in this study. The method estimates values in unsampled locations based on the moving average of the interest variables, satisfying various distribution forms of data. Ordinary kriging is a spatial estimation method that provides a minimum error-variance estimate of any unsampled value. The

correct variogram in kriging interpolation can guarantee the accuracy of the interpolation results. The accuracy of the spatial interpolation analysis is verified by comparing the difference between the measured values and the predicted values to select the best variogram model. In order to test the variogram model, the cross-validation method is used to determine whether the parameters of the variogram model are correct (Fig. 7). The distribution of the residual is normal and the mean of error between the actual and estimated Cu grade values is equal to 0 (Table 1). This result indicates that this model is reasonable and that the variogram parameters used for estimating the Cu grade are unbiased.